# Tendency in Pulmonary Aspergillosis Investigation during the COVID-19 Era: What Is Changing?

**DOI:** 10.3390/ijerph19127079

**Published:** 2022-06-09

**Authors:** Giuseppina Caggiano, Francesca Apollonio, Mila Consiglio, Valentina Gasparre, Paolo Trerotoli, Giusy Diella, Marco Lopuzzo, Francesco Triggiano, Stefania Stolfa, Adriana Mosca, Maria Teresa Montagna

**Affiliations:** 1Interdisciplinary Department of Medicine, Hygiene Section, University of Bari Aldo Moro, Piazza G. Cesare 11, 70124 Bari, Italy; francesca.apollonio@uniba.it (F.A.); paolo.trerotoli@uniba.it (P.T.); giusy.diella@uniba.it (G.D.); stolfastefania@gmail.com (S.S.); adriana.mosca@uniba.it (A.M.); mariateresa.montagna@uniba.it (M.T.M.); 2Department of Biomedical Science and Human Oncology, University of Bari Aldo Moro, Piazza G. Cesare 11, 70124 Bari, Italy; mila.consiglio@uniba.it (M.C.); valentina.gasparre@uniba.it (V.G.); marco.lopuzzo@uniba.it (M.L.); francesco.triggiano@uniba.it (F.T.)

**Keywords:** aspergillosis, CAPA, invasive pulmonary aspergillosis, galactomannan, 1,3-β-D-glucan

## Abstract

Aspergillosis is a disease caused by *Aspergillus*, and invasive pulmonary aspergillosis (IPA) is the most common invasive fungal infection leading to death in severely immuno-compromised patients. The literature reports *Aspergillus* co-infections in patients with COVID-19 (CAPA). Diagnosing CAPA clinically is complex since the symptoms are non-specific, and performing a bronchoscopy is difficult. Generally, the microbiological diagnosis of aspergillosis is based on cultural methods and on searching for the circulating antigens galactomannan and 1,3-β-D-glucan in the bronchoalveolar lavage fluid (bGM) or serum (sGM). In this study, to verify whether the COVID-19 period has stimulated clinicians to pay greater attention to IPA in patients with respiratory tract infections, we evaluated the number of requests for GM-Ag research and the number of positive tests found during the pre-COVID-19 and COVID-19 periods. Our data show a significant upward trend in GM-Ag requests and positivity from the pre-COVID to COVID period, which is attributable in particular to the increase in IPA risk factors as a complication of COVID-19. In the COVID period, parallel to the increase in requests, the number of positive tests for GM-Ag also increased, going from 2.5% in the first period of 2020 to 12.3% in the first period of 2021.

## 1. Introduction

Aspergillosis is a disease caused by *Aspergillus*, a common mold that lives indoors and outdoors. Most people routinely inhale *Aspergillus* spores without getting sick. Of the hundreds of *Aspergillus* species that exist in nature, only a few can cause infection in patients. Among these, *A. fumigatus* is the most frequent cause of infection, accounting for over 90% of infections, followed by *A. flavus* and *A. niger* [1].

Immunosuppressed subjects or those with severely debilitating diseases can develop a wide spectrum of fungal diseases that contribute to morbidity and mortality [2,3,4,5,6]. The main ones are SAFS (severe asthma with fungal sensitization), ABPA (allergic bronchial pulmonary aspergillosis), CPA (chronic pulmonary aspergillosis), IBA (invasive bronchial aspergillosis), and IPA (invasive pulmonary aspergillosis) [7].

SAFS is a form of steroid-refractory asthma with evidence of sensitization to *Aspergillus* or to other fungal allergens without clinical and radiographic signs [8].

ABPA is an allergic lung disease related to *A. fumigatus* hypersensitivity and is the most severe form of aspergillosis that develops in particular subsets of atopic patients, such as individuals with cystic fibrosis [9,10] or a genetic predisposition for ABPA [11,12].

CPA defines the entire spectrum of non-invasive diseases caused by *Aspergillus*, including aspergilloma, which is generally seen in residual tuberculosis cavities [13].

IBA is a form of invasive bronchial aspergillosis that is often acquired within 3 months of a lung transplantation that manifests with endobronchial plaques, nodules or areas of ulceration and necrosis that may extend to the adjacent pulmonary parenchyma and vasculature [14].

IPA is the most common invasive fungal infection that leads to death in severely immuno-compromised patients, especially in recipients of hematopoietic stem cell transplantation (HSCTs) and solid-organ transplants (accounting for 43% and 59% of all deaths, respectively). It mostly occurs late (40 to 80 days) or very late (>80 days) after engraftment [15,16,17]. IPA has now become the most expensive fungal disease in hospital settings because of its prevalence and costly treatments [18]. It is included among the fungal superinfections occurring in patients with viral pneumonia. In fact, respiratory viruses cause direct damage to the airway epithelium, favoring a state of immunosuppression due to lymphopenia (reduced activity of macrophages and natural killer cells) and enabling *Aspergillus* to invade tissues [19,20].

In 2012, the H1N1 virus caused a severe influenza A infection, which, in many patients, worsened due to its association with *Aspergillus* infection. Today, it is known that Influenza-associated pulmonary aspergillosis (IAPA) complicates the clinical course of critical patients with acute respiratory distress syndrome (ARDS) [21,22]. Some authors report that IAPA is present in between 17% and 29% of severe influenza patients and has a high mortality rate (67%) [23,24].

In December 2019, COVID-19 emerged from Wuhan, China, and has since become a global pandemic [25]. Since then, several studies have shown that COVID-19 patients are at an increased risk of secondary infections, and many microorganisms have been detected as possible co-pathogens, giving rise to another serious concern in their treatment [26,27,28,29,30].

The literature reports fungal co-infections in COVID-19 patients [31,32,33,34]. In addition, corticosteroid therapy is an immunosuppressive factor related to IPA that has been used in up to 46% of critical COVID-19 patients [35]. In particular, cases of COVID-19-associated pulmonary aspergillosis (CAPA) infections are on the rise and are associated with high morbidity and mortality [36,37,38,39,40,41]. Diagnosing CAPA clinically is complex since the symptoms are non-specific, performing a bronchoscopy is difficult, and imaging tools may not differentiate between COVID-19 pneumoniae and invasive aspergillosis.

In the pandemic era, these considerations have led physicians to pay greater attention to *Aspergillus* complications in other types of patients, in addition to those at high risk of mycotic complications (e.g., hematological complications), leading to the detection of an infectious risk that is often underestimated [3,15]. Generally, for the microbiological diagnosis of aspergillosis, bronchoalveolar lavage fluid and lung biopsy samples are the specimens of choice. Diagnosis is mainly based on cultural methods and a search for circulating antigens (galactomannan and 1,3-β-D-glucan in the bronchoalveolar lavage fluid (bGM) and serum (sGM)). It is believed that these biomarkers are detectable even before the onset of clinical symptoms and hence are useful for guiding pre-emptive treatment. In particular, GM is the carbohydrate constituent of the cell wall of *Aspergillus* genus; its identification in bronchoalveolar lavage fluid (BAL) is highly indicative of IPA, as the antigen is released during the active growth of the fungus [42,43]. Levels of bGM are always higher than those of sGM [44]. Consequently, bGM is considered the primary laboratory test for diagnosing secondary IPA in patients with a severe viral infection.

Direct inspection of the trachea and bronchi via a bronchoscopy allows for the identification of patients with pulmonary aspergillosis [38], especially in subjects negative for SARS-CoV-2. In COVID-19 patients, the reduced use of this diagnostic method, in order to protect healthcare workers from exposure to aerosols, and the low detection sensitivity of galactomannan in serum limit the diagnosis of invasive airway aspergillosis [45,46].

The most recent 2020 consensus guidelines [47] aimed at establishing the definition of an invasive fungal infection recognize two positive PCR tests as having sufficient specificity to confirm the diagnosis of aspergillosis. However, this consensus does not include ICU patients. Consequently, the *Aspergillus* PCR and the 1-3-β-D-glucan test, as mycological evidence, would not be recommended for the diagnosis of IPA in these patients [48]. Thus, to improve patient management, routine screening of respiratory specimens for *Aspergillus* using GM assays from serum or BAL is essential.

The aim of this study is to verify whether the COVID-19 period stimulated clinicians to pay greater attention to IPA in patients with respiratory tract infections. We therefore evaluated the number of requests for GM-Ag tests and the number of positive tests found during the pre-COVID-19 and COVID-19 periods.

## 2. Materials and Methods

The study was conducted in a large hospital in southern Italy, which has 1550 beds and is divided into 32 pavilions that house medical and surgical units. A central Microbiology and Virology Laboratory receives all the biological samples for different examination. GM-Ag research requests from each department that had admitted patients with bronchopulmonary diseases were investigated.

The biological samples were delivered to the central microbiology laboratory. This was equipped with a computerized system that assigned a unique identification code to each biological sample and stores the data. In addition, each analysis performed on the samples corresponded to a code of the health service.

The data were extracted using the codes of the health service without accessing the patients’ sensitive data. This information was grouped bimonthly, identifying January 2019–February 2020 as the pre-COVID period and March 2020–June 2021 as the COVID-19 period.

GM-Ag tests on bronchoalveolar fluid (BAL) or serum were performed by immunosorbent assays linked to a sandwich enzyme (PlateliaTM Aspergillus EIA, Biorad Laboratories) according to the manufacturer’s instructions. The result was rated positive if the positivity index was >1.

### Statistical Analysis

The data were summarized as counts and percentages. Comparisons between independent groups were performed using the chi-squared test. A Cochran–Armitage test was performed to evaluate the trends in the percentages of positive results for bimonthly periods. Both univariable and multivariable logistic regression models were performed to assess the probability of a positive outcome by ward to adjust the results for age and gender. The bimonthly data were added to the multivariable logistic regression model to evaluate the trends by clinical unit.

The data were analyzed using SAS/STAT® Statistics version 9.4 (SAS Institute, Cary, NC, USA) for personal computers. For statistical significance, a *p*-value of less than 0.05 was considered.

## 3. Results

In total, from January 2019 to June 2021, 7162 GM-Ag tests were included in our data, of which 1902 (26.5%) were taken during the pre-COVID period and 5260 (73.4%) were taken during the COVID period.

In the pre-COVID period (January 2019–February 2020), the number of GM-Ag tests was more or less constant with an average of 280 requests every two months.

On the contrary, in the COVID period, although there were no significant differences from March to August, the temporal trend of requests progressively increased from 278 (5.3%; 278/5260) in March–April 2020 to 529 (10%; 529/5260) in September–October 2020 and to 1538 (29.2%; 1538/5260) in March–April 2021 (Figure 1).

In the period September–October 2020, the temporal trend increased by 2.1% compared to the same period in the previous year (September–October 2019; 278 vs. 529 requests; 5.3% vs. 7.4%) In the subsequent months, requests decreased (739 requests; 10.4%) (Figure 1).

The trend of positivity in the pre-COVID period exhibited random fluctuations with higher values in the period January–February 2019 (4.9%) but without a spread or a slow growth trend.

In the COVID period, parallel to the increase in requests, the number of positive tests for GM-Ag also increased, going from 2.5% in May–June 2020 to 5.6% in November–December 2020, with a spread in January–February 2021 (12.3%) (Cochrane–Armitage = 6.79; *p* <0.0001).

The GM-Ag requests came from the following departments: COVID (*n* = 2856; 39.8%), Onco-haematology (*n* = 2758; 38.5%), Internal Medicine (*n* = 533; 7.4%), Pediatric Oncohaematology (*n* = 195; 2.7%), ICU (*n* = 165; 2.3%), Infectious Diseases (*n* = 131; 1.8%), Pneumology (*n* = 87; 1.2%), Surgery (*n* = 67; 0.9%), Other Specialist (*n* = 220; 3.1%), Outpatient Clinic (*n* = 46; 0.6%), and Outpatient Hospital (*n* = 104; 14%). A comparison between the pre-COVID and COVID periods shows a statistically significant variation in the number of requests (64 vs. 469) coming from the Internal Medicine department (3.4% vs. 8.9%; *p* < 0.0001), while the difference in the other departments was not statistically significant.

The percentage of positive results increased from 2.8% in the pre-COVID period to 6.1% in the COVID period (chi-square 32.71; *p* < 0.0001), even after adjusting for department (*p* = 0.0004). Positive results were not homogeneous between departments (*p* < 0.01) (Table 1).

A difference in the percentage of positive results between the pre-COVID and COVID periods was observed in all departments. The departments with remarkable increases include Pediatric Oncohematology (rose from 1.2% to 6.4%), Pneumology (rose from 2% to 16.2%), and Surgery (rose from 10% to 21.3%).

The logistic regression allowed us to evaluate the trends when adjusting for the age and sex of the subjects as well as the effect of department. The interaction terms in the multivariable regression were removed because they were not statistically significant. In contrast, both in the univariable and multivariable models, department and age (entered in the model as classes) had statistically significant effects. In the multivariable analysis (Table 2), all departments showed a statistically significantly higher risk with respect to the Oncohematology Unit, which was chosen as a reference because during the pandemic period it had the lowest percentage of positive test results. The model showed that the Pediatric Oncohematology (OR = 14.04; 95% CI 3.82–51.62) and ICU (OR = 6.38; 95% CI 3.94–10.32) departments had the highest probabilities of returning a positive test. Outpatients had an odds ratio (OR) of 7.19 (95% CI 3.32–15.59), which was an unexpected result.

## 4. Discussion

IPA infection is emerging as a serious complication in patients with viral pneumonia. Some studies have shown that the mortality rates of these patients may be 16% or even 25% higher compared to those of patients with no evidence of aspergillosis [46,49]. These excess mortality rates are similar to those seen for IAPA patients, where the ICU survival rate has been found to be 24% lower than in patients without aspergillosis [22].

There are obvious similarities between IAPA and CAPA, including high prevalence, absence of classic host factors for invasive fungal infection, similar timing in disease diagnosis after ICU admission, and the presence of lymphopenia. However, it is still not entirely clear whether COVID-19 is the main risk factor for CAPA or whether other risk factors, such as corticosteroid therapy, age, comorbidities (e.g., diabetes, asthma, COPD, cancer), and, potentially, male sex, further increase the risk of disease progression [47,50,51,52,53].

Among ventilated COVID-19 patients, preliminary studies have reported a high incidence of invasive aspergillosis, which may affect up to 30% of intubated patients [36,38,54]. During the same period, Bartoletti et al. [46] published a prospective multicenter cohort study conducted in adult patients with laboratory-confirmed SARS-CoV-2 infections who required admission to the ICU for mechanical ventilation. This study highlights the high incidence of CAPA in these patients and a statistically significant increase in mortality (*p* = 0.04), which is likely related to the use of corticosteroid and tocilizumab in the treatment of COVID-19. CAPA was defined as meeting at least one of the following criteria: sGM index > 0.5, bGM index > 1.0, isolation of *Aspergillus* in BAL. This study also reports a significant association between bGM index at ICU admission and mortality at 30 days (*p* = 0.014). When using bGM as a diagnostic benchmark for CAPA, sGM had 6% specificity, 34% sensitivity, 19% positive values, and 49% negative predictive values [46].

Our study found that GM-Ag demand increased significantly during the COVID-19 period compared to the pre-COVID-19 period (5260 vs. 1902, respectively) (*p* < 0.00001), likely because patients with lung diseases need a differential diagnosis between SARS-CoV-2 infection and other infectious complications, including aspergillosis. This is supported by the fact that the greatest number of requests came from the COVID-19 and Internal Medicine Units. The Internal Medicine ward often admits post-COVID-19 patients when they have not fully overcome the consequences of the disease but the swab test for SARS-CoV-2 has turned negative. This could explain why a significant difference in requests between before and during COVID-19 was observed only in the Internal Medicine ward. In the Oncohematology, Infectious Diseases, Pneumology, and remaining wards, both the number of requests and the positive result for GM-Ag did not show substantial differences between the pre-COVID-19 and COVID-19 periods. On the other hand, there was an evident increase in positive GM-Ag tests in the ICU, Pneumology department, Surgery department, and Outpatient Clinic.

Concerning requests, the higher number in the last quarter of 2020 can likely be explained by the second wave of the pandemic in Italy, which coincided with the first in Puglia.

It is important to highlight that the decrease in positive tests for GM-Ag that emerged in the second quarter of 2021 may be attributable to the introduction of the National Guidelines on COVID-19 Therapy, which led to less use of corticosteroids. In fact, the diagnosis of CAPA requires a strong correlation between clinical, radiographic, and laboratory data, since the growth of *Aspergillus* within biological material or a positive test for fungal biomarkers does not necessarily indicate invasive disease [38]. In this study, the cultural investigations of BAL were not considered, although the research of bGM was always accompanied by the cultural investigation necessary, in case of positivity, to identify the species and evaluate the anti-mycotic susceptibility.

Clinical vigilance is recommended in patients with critical health conditions, even without the classical host criteria. In these cases, thorough teamwork, an accurate diagnostic work-up, and early initiation of therapy can save lives [35].

Therefore, patients admitted to hospitals with respiratory viral infections, particularly COVID-19, should be systematically screened for fungal complications in order to reduce the negative consequences of such colonization.

Our study presents some limitations. It was conducted on the requests received by the clinicians, so the frequency of positive results could be biased by the non-random selection of patients submitted to the diagnosis; neither matching was applied (i.e., by clinical unit, comorbidities, age and sex as criteria). Some of these data were not available in the database of the laboratory. The attention of clinicians to these infectious events has increased in time, showing more cases than expected even in units considered at lower risk, but this is related to the increase in laboratory confirmation tests. It is already known that the risk related to particular units, such as Intensive Care, could be considerable, and the increase in attention could have led to rise in requests and positivity, but this could bias the conclusions on increase prevalence.

## 5. Conclusions

The data in this study show a significant upward trend in GM-Ag requests and positivity from the pre-COVID to the COVID period, which is attributable to the common respiratory target of SARS-CoV-2 and *Aspergillus* spp. and to the increase in IPA risk factors as a complication of COVID-19. Immunosuppression, non-invasive ventilation, oro-tracheal intubation, and the use of high-dose corticosteroid therapy for long periods make patients more susceptible to infection, leading clinicians to seek further diagnostic investigations.

COVID-19 has highlighted the need to promote primary prevention to avoid serious infectious complications and to define adequate diagnostic–therapeutic protocols in order to reduce the incidence of CAPA.

Currently, the data and patient classification should be interpreted with caution until more scientific evidence becomes available. High-risk patient groups need to be better defined to include those undergoing corticosteroid treatment, which seems to be an important risk factor for CAPA.

## Figures and Tables

**Figure 1 ijerph-19-07079-f001:**
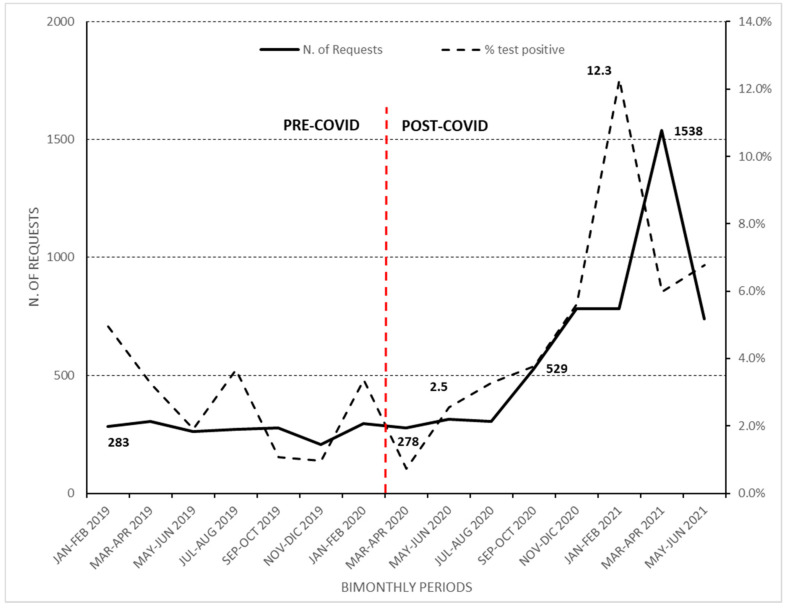
Distribution of the number of requests for GM-Ag tests and percentage of positive results at bimonthly intervals. January 2019–February 2020 was identified as the pre-COVID period; March 2020–June 2021 as the COVID-19 period.

**Table 1 ijerph-19-07079-t001:** Number of tests requested for GM-Ag and positive tests (%) in the pre-COVID and COVID periods by department. January 2019–February 2020 was identified as the pre-COVID period; March 2020–June 2021 as the COVID-19 period.

	Pre-COVID Period		COVID Period		Total	
Departments	No. of Positive Tests/No. of Total Tests	Positive Tests (%)	No. of Positive Tests/No. of Total Tests	Positive Tests (%)	No. of Positive Tests/No. of Total Tests	Positive Tests (%)
Oncohematology	22/1428	1.5	48/1330	3.6	70/2758	2.5
Pediatric Oncohematology	1/86	1.2	7/109	6.4	8/195	4.1
Internal Medicine	4/64	6.2	37/469	7.9	41/533	7.7
ICU	11/39	28.2	15/126	11.9	26/165	15.7
Pneumology	1/50	2.0	6/37	16.2	7/87	8
COVID	/	/	166/2856	5.8	166/2856	5.8
Outpatients	2/12	16.7	6/34	17.6	8/46	17.4
Infectious Diseases	2/73	2.7	4/58	6.9	6/131	4.5
Surgery	2/20	10.0	10/47	21.3	12/67	17.9
Other Specialties	6/89	6.7	12/131	9.2	18/220	8.2
Outpatient Hospital	3/41	1.5	11/63	17.5	14/104	12.5
Total	54/1902	2.8	322/5260	6.1	376/7162	5.2

**Table 2 ijerph-19-07079-t002:** Odds ratios (ORs) and 95% confidence intervals determined by logistic regression.

		Univariable Analysis		Multivariable Analysis	
		OR	95% OR	OR	95% OR
**Gender**	Female	0.95	0.77–1.18	0.98	0.79–1.22
	Male	reference		reference	
**Age class**	0–18	reference		reference	
	19–34	1.51	0.54–4.16	8.94	2.12–37.66
	35–49	2.48	0.99–6.24	17.79	3.81–83.04
	50–64	1.82	0.73–4.52	11.64	2.51–53.94
	65 or over	3.01	1.23–7.39	17.62	3.83–81.12
**Departments**	Oncohematology	reference		reference	
	Pediatric Oncohematology	1.59	0.75–3.34	14.04	3.82–51.62
	Internal Medicine	3.48	2.38–5.1	2.4	1.6–3.59
	ICU	7.76	4.88–12.32	6.38	3.94–10.32
	Pneumology	4.29	2.07–8.87	3.96	1.89–8.28
	COVID	2.5	1.89–3.31	1.56	1.14–2.15
	Outpatients	9.01	4.21–19.31	7.19	3.32–15.59
	Infectious Diseases	2.35	1.11–4.98	2.35	1.1–5.02
	Surgery	7.56	3.9–14.65	5.42	2.75–10.69
	Other Specialties	3.35	1.96–5.72	2.91	1.69–5
	Other Hospital	6.01	3.26–11.06	5.89	3.17–10.94
**Bimonthly trend**	for each period	1.11	1.08–1.14	1.09	1.06–1.13

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
