# Peer review of "Tendency in Pulmonary Aspergillosis Investigation during the COVID-19 Era: What Is Changing?"

_ijerph, 2022, doi:10.3390/ijerph19127079_

Round 1
Reviewer 1 Report
This work is a comparative study of the results of the detection of Aspergillus fumigatus galactonannan antigen in bronchoalveolar lavage and serum of patients from a high complexity hospital in southern Italy. The comparison was made between the pre-Covid 19 era and during this virus pandemia. The study is well planned and developed and the conclusions are justified by the data obtained. The presentation is clare.
Your presentation is correct; I have only two minor objections, which do not alter the paper quality: the introdution is too long and no comment are made on the results of bronchoalveolar lavage cultures, althogh, these were not the objetive of this study, some comment could be made in the discussion.
Author Response
We thank the Reviewer for the kinds comments that permitted us to improve our paper.
- We tried to reduce the introduction, but this would involve the elimination of some important points to define the topic of manuscript
- We added a comment about bronchoalveolar lavage cultures in the discussion paragraph (line 242-244)
Reviewer 2 Report
Thank you for giving me the opportunity to review this manuscript. This paper report the increasing number of GM-Ag request and positive results during the COVID-19 era in departments managing COVID-19 patients. The paper is well written.
However, I have a couple of comment :
1) The method section is insuffisant. Please characterise your study, the collected data...
2) Data concerning the patients as for instance whether they were received corticosteroids or whether that had a confirmed disease could have added much to the paper.
3) Please add a paragraph on the limitation of your work.
4) I think the title does not suit much. Actually with such a title we expect much more but this paper relates on fungal biomarker prescription during the COVID-19 era. Could you think of another title ?
Author Response
We thank the Reviewer for comments that permitted us to improve our paper.
- The method section is insuffisant. Please characterise your study, the collected data...
We thank the Reviewer for this observation. We added a comment in the Materials and Methods paragraph (lines 112-119)
- Data concerning the patients as for instance whether they were received corticosteroids or whether that had a confirmed disease could have added much to the paper.
We agree to Reviewer, but we did not consider the patients’ data, so we didn't need the approval of the ethics committee. The study topic was to verify, through number of requests for GM-Ag tests- whether the COVID-19 period stimulated clinicians to pay greater attention to IPA in patients with respiratory tract infections
- Please add a paragraph on the limitation of your work.
We added the work limitation in the paragraph conclusion 251- 260
- I think the title does not suit much. Actually with such a title we expect much more but this paper relates on fungal biomarker prescription during the COVID-19 era. Could you think of another title ?
Agree to the Reviewer, we modified the title “Tendency in Pulmonary Aspergillosis investigation during the COVID-19 Era: What is Changing?”
